# Reduction in Spoilage Microbiota and Cyclopiazonic Acid Mycotoxin with Chestnut Extract Enriched Chitosan Packaging: Stability of Inoculated Gouda Cheese

**DOI:** 10.3390/foods9111645

**Published:** 2020-11-11

**Authors:** Kristi Kõrge, Helena Šeme, Marijan Bajić, Blaž Likozar, Uroš Novak

**Affiliations:** 1Department of Catalysis and Chemical Reaction Engineering, National Institute of Chemistry, Hajdrihova 19, 1000 Ljubljana, Slovenia; kristi.korge@ki.si (K.K.); marijan.bajic@ki.si (M.B.); blaz.likozar@ki.si (B.L.); 2Department of Chemistry and Biotechnology, Tallinn University of Technology, Akadeemia tee 15, 12618 Tallinn, Estonia; 3Acies Bio d.o.o., Tehnološki park 21, 1000 Ljubljana, Slovenia; helena.seme@aciesbio.com

**Keywords:** active chitosan-based film, chestnut extract, gouda inoculated cheese, cyclopiazonic acid, tannins, antimicrobial and antifungal food packaging, biopolymers

## Abstract

Active chitosan-based films, blended with fibrous chestnut (*Castanea sativa* Mill.) tannin-rich extract were used to pack Gouda cheese that has been contaminated with spoilage microflora *Pseudomonas fluorescens*, *Escherichia coli*, and fungi *Penicillium commune*. A comprehensive experimental plan including active chitosan-based films with (i) chestnut extract (CE), (ii) tannic acid (TA), and (iii) without additives was applied to evaluate the film′s effect on induced microbiological spoilage reduction and chemical indices of commercial Gouda cheese during 37 days while stored at 4 °C and 25 °C, respectively. The cheese underwent microbiology analysis and chemical assessments of ultra-high-performance liquid chromatography (UHPLC) (cyclopiazonic acid), pH, and moisture content. The biopackaging used for packing cheese was characterized by mechanical properties before food packaging and analyzed with the same chemical analysis. The cheese microbiology showed that the bacterial counts were most efficiently decreased by the film without additives. However, active films with CE and TA were more effective as they did not break down around the cheese and showed protective properties against mycotoxin, moisture loss, and pH changes. Films themselves, when next to high-fat content food, changed their pH to less acidic, acted as absorbers, and degraded without plant-derived additives.

## 1. Introduction

In chemical attributes, the main macromolecules in Gouda cheese are lipids (24%) followed by proteins. With the nutritious media, this semihard cheese can be highly susceptible to microbial hazard in ambient conditions (room temperature, high O_2_), including the mold’s ability to stay vital at refrigeration temperature, but also at low values of O_2_, pH, and water activity [1]. The water activity of the Gouda cheese itself has been reported as an average of 0.972 [2]. When contaminated with pathogenic bacteria or toxigenic fungi, a rapid growth fosters deterioration of the texture and mycotoxin production [3,4]. There is a list of cheese contaminators, but a few concerning ones that can be named are bacteria *Pseudomonas fluorescens* and *Escherichia coli* and fungi *Penicillium commune* [5,6]. The latter is known for producing mycotoxin cyclopiazonic acid (CPA), strongly implicated as a causative agent in mycotoxicosis for animals and is potentially harmful to humans [7,8,9]. For these reasons, various types of cheese require a different kind of packaging concepts that meet the demand for prolonged shelf life.

Within current technologies to protect cheese from spoilage, it is either paraffined or packed in flexible film, including conventional polyethylene (PE) or similar polymer materials such as polyethylene terephthalate (PET) or polypropylene (PP) [10]. Both materials represent good barrier properties, although PE (or similar) material act as a single-use packaging and thus in most cases ends up as a non-degradable waste. The wax layer favors bacterial growth, causing off-flavors and gas formation when not applied to the cheese surface properly [2]. On the other hand, films using chitosan constitute thin layers of materials that have been successfully tested to substitute synthetic packaging and prolong the shelf life of cheese in regard to safety [11,12,13]. Numerous natural substances have been tested and interpreted through modeling to design applicable biomaterials [14,15]. However, to our knowledge, none of the created compositions take into account adjacent food macromolecular composition and its characteristic microbial population effect on the films themselves [16]. Therefore, additionally to food analysis in the biopackaging, it is necessary to monitor how the films are changed in times when they are in contact with certain food macromolecules/microbes. The simultaneous analysis of two matrices is required input to the ongoing engineering of the films, which makes choosing suitable storage material for food more efficient.

Engineering of the bio-based film starts with the matrix-forming biomaterial. Chitosan is a long-chain polymer with reactive OH^−^ and β-(1–4) positioned NH_2_^+^ groups, mainly chemically converted biomaterial from chitin, and used as one among others [17]. Due to the conversion, it receives higher solubility in a mildly acetic aqueous solution, which is vital for the film-forming solution (FFS) homogeneity. By being able to aggregate with negatively charged molecules of fats (oleic, linoleic, palmitic, stearic, linolenic), it performs antihyperlipidemic action through ionic complexes (between NH_2_^+^ and O^−^) [18]. In the form of film, it bestows antimicrobial properties through either electrostatic interaction with the cell wall by changing the cell permeability, or metal chelation with the outcome of collapsing/distorting the outer membrane. In the end, the DNA of bacteria will be damaged and depleted [19]. The biopolymer deacetylation (DA) level, molecular weight, and pH are essential factors to the antimicrobial activity [20,21]. In fact, values of DA and molecular weight (M_w_) are in correlation. The high numeric value of these two parameters gives chitosan enhanced binding affinity and uptake capacity; employs chitosan into a non-degradable, less penetrable film matrix formation, which is essential when preserving food and plays a role when applying against certain spoilage bacteria. Furthermore, since chitosan amine groups are becoming ionized at pH < 6, antimicrobial activity improves at low pH [19].

The above-mentioned properties of chitosan are well-aligned with properties of plant derivative extracts, frequently used as active additives to engineered biofilms [22]. In this regard, the chestnut extract is one of the widespread derivatives that is also known for enhancing the film’s permeability properties [23,24]. Various plant segments (fruits, leaves, galls, bark, and wood) are used for the extract. It is a multifunctional component mainly consisting of starch (40–60%), followed by condensed tannins with flavonoid core and, most importantly, hydrolyzable tannins (HTs). HTs are a mix of simple phenols, which also have the coloring effect. Their chemistry is broad, covering interactions not only with proteins but several other organic nitrogen compounds, including arginine, chitin, and chitosan. The higher reactivity is related to sufficient amine groups and higher M_w_ [25]. Studies report chestnut extract tannins interactions with abdominal cholesterol and thus lowering adiposity in mice [26], a decrease in *Cladosporium cladosporioides* on sheep cheese rind [7], *L. monocytogenes* in Emmental cheese [27], *E. coli*, *P. fluorescens* in mozzarella cheese [28], and reduction in mycotoxins [29]. Yet even though the chestnut extract is widely spread, there seems to be a gap in knowledge of how the component combines with chitosan to prevent food spoilage.

This work aimed to prepare new food packaging material, as in chitosan-based films with chestnut extract, to enhance the chitosan-based film’s antimicrobial activity by incorporating active HTs from chestnut extract (CE). For better realization of the activity, the films were coupled with reference (tannic acid:chitosan (TA:CH) and CH) films. Enhanced film′s antimicrobial activity was tested on induced Gouda cheese spoilage, chosen to study the impact on high lipid food. To do so, comparative analysis of microbiology, ultra-high-performance liquid chromatography (UHPLC), pH and moisture mobility were applied in two conditions (4 °C and 25 °C) for 37 days. To our knowledge, there are no reports on CPA measurements in biopackaging, which was one of the objectives of this study. Biopackaging changes and food changes are described and presented as a two-way system.

## 2. Materials and Methods

### 2.1. Materials

High molecular weight chitosan (CH) (acetylation degree ≥ 75%, 310–375 kDa), lactic acid (LA) (purity ≥ 85%), and tannic acid (TA) were purchased from Sigma-Aldrich (Steinheim, Germany), while methanol and acetonitrile were purchased from Avantor Performance Materials (Gliwice, Poland) and Honeywell (Hannover, Germany), respectively. Sodium dihydrogen phosphate dihydrate and ortho-phosphoric acid (purity ≥ 85%) were obtained from Merck (Darmstadt, Germany), ammonium acetate from Kemika (Zagreb, Croatia), and glycerol (GLY) from Pharmachem Sušnik (Ljubljana, Slovenia). Commercially available CE (≥75% tannins; <4% of ash) was provided by the company Tanin Sevnica (Sevnica, Slovenia). All chemicals except LA were of analytical grade. According to EU legislations (EU) 2017/2470 (chitosan) 2017, (EU) 2017/66 (tannic acid) 2016, (EU) no 231/2012 (lactic acid and glycerol) 2012 [30] chestnut extract specification (provided by producer company), all the substances can be considered as food additives. Milli-Q^®^ water was used throughout all the experiments.

### 2.2. Film-Forming Solutions and Chitosan-Based Films

#### 2.2.1. Film-Forming Solutions

All FFSs formulations were prepared at ambient conditions by adding predetermined amounts of CH (% *w*/*v*) and GLY (% *w*/*w*, calculated per mass of CH) in the solvent (1% (*v*/*v*) aqueous solution of lactic acid) followed by continuous stirring (1000 rpm; 12 h; room temperature, 24 °C) on RCT magnetic stirrer (IKA, Staufen, Germany) [31]. The predetermined amounts of CE or TA were added subsequently after the mixing step, and the mixtures were homogenized (6000 rpm; 2 min) on Ultra-Turrax^®^ T50 (IKA) and left overnight to get rid of the air bubbles formed during this process. A small amount of stable foam that was formed on the top of the mixtures was eventually removed by using a laboratory spatula.

#### 2.2.2. Chitosan-Based Films

Prepared FFSs were cast in polyurethane Petri dishes (approximately 0.32 mL/cm) and left under constant airflow box (Microbium d.o.o, Ljubljana, Slovenia) at room temperature 24 ± 2 °C for the next 24 h. Obtained films were peeled off from rectangular Petri dishes (12 cm × 12 cm), and stored in an airtight container (24 °C, no exposure to light) until further analysis.

### 2.3. Gouda Cheese Preparation

The Gouda cheese was purchased from a local supermarket in Slovenia, Ljubljana. Before repacking, the cheese blocks were cut under the sterile constant airflow box into uniform pieces with an average weight of 22 g.

### 2.4. Experimental Design

A 4:4:2 factorial experiment design (4 different packaging sets × 4 time points × 2 temperatures) was implemented during this study (Scheme 1). Accordingly, three sets of chitosan-based films, namely chestnut extract:chitosan (CE:CH), tannic acid:chitosan (TA:CH), and blank chitosan (CH) with extra layer of polyamide:polyethylene (PA:PE) vacuum bags (Status d.o.o Metlika, Slovenia) were prepared. A reference set with PA:PE was prepared next to biopackaging sets.

The precut cheese was placed onto a Petri dish, inoculated with chosen spoilage microorganisms (*P. fluorescens*, *E. coli*, *P. commune*), and packed into a sachet of one 12 cm × 12 cm sheet of chitosan film. Each sachet was prepared by heat sealing (165 °C, 700 Pa, 7 s) on HST-H6 heat seal tester PARAM^®^ (Labthink, Jinan, China) along the long and then short edge. After the insertion of the spoiled food, the sachet was heat-sealed once more from a short open edge. The same process was completed with the unspoiled food and all the procedures were conducted under the sterile conditions of a constant airflow box. All the cheese:biofilms sample sets were additionally packed into an extra layer of PA:PE vacuum bags and vacuum sealed. This was achieved using a vacuum sealer (Status d.o.o, Metlika, Slovenia) with a purpose of preventing environmental effects and to study the ultimate mutual impact of the matrices inside the sets. The named sample sets with spoilage bacteria were stored in 4 °C conditions while samples with spoilage fungi in environmental conditions of 4 °C and 25 °C for 37 days (day 0, 7, 14 and 37). The full name of the samples and their abbreviations are presented in Table 1.

#### 2.4.1. Bacterial Inoculation

*E. coli* K12 and *P. fluorescens* NRRL B-253 were grown overnight at 30 °C in 2x YT medium. The medium was discarded, and the culture was resuspended in sterile saline solution. Cheese bricks were aseptically cut to a rectangular dimension of 50 mm × 25 mm × 8 mm (~22 g). Then, 100 µL of bacterial inoculum (*E. coli* 6.5 log_10_ CFU/g and *P. fluorescens* 8.3 log_10_ CFU/g) was spread onto the surface of the cheese slice and covered with biofilm or in case of control, no biofilm was used. The sachet was put into PA:PE bag and vacuum-sealed (Status d.o.o, Metlika, Slovenia) and stored at 4 °C.

#### 2.4.2. Fungal Inoculation

*P. commune* NRRL 894 was grown on malt extract agar plates for 10 days to obtain spores. Spores from the plate were collected in sterile saline solution and homogenized with vortexing. Then, 100 µL (4.3 log_10_ CFU/g) of fungal inoculum was spread onto the surface of the cheese slice and covered with chitosan film, or in case of control, only a vacuum bag was used. The packet was put into PA:PE bag and vacuum-sealed (Status d.o.o, Metlika, Slovenia) and stored at 4 °C or 25 °C.

### 2.5. Microbiological Analysis

On days 0, 7, 14, and 37, each cheese sample was opened aseptically and cut in half. One half of the sample (approximately 10 g) was mixed with 90 mL of sterile saline solution and homogenized with a stomacher (Lawson Scientific, Ningbo, China). The suspensions were appropriately diluted in sterile saline solution and plated on selective medium. The 2x YT medium with incubation at 37 °C for 24 h was used for *E. coli* count, Pseudomonas selective agar and incubation at 30 °C for 48 h was used for *P. fluorescens* count, and malt extract agar with incubation at 25 °C for 5 days was used for *P. commune* enumeration.

### 2.6. Chemical and Physical Analysis

#### 2.6.1. UHPLC

Analysis of the liquid samples was performed by using ultra-high-performance liquid chromatography—UHPLC (Thermo-Fisher Scientific UltiMate™ 3000, Waltham, MA USA)—equipped with a 5.0 μm; 4.6 × 150 mm Hypersil GOLDTM Amino column (ThermoFisher Scientific, Waltham, MA USA), heated to 30 °C, and equipped with a DAD detector. Cyclopiazonic acid was identified by retention time and UV-Vis spectra comparison to reference standards using UV–VIS spectra between 282–283 nm. The compound was quantified by external calibration standards. The mobile phase with a flow rate of 0.6 mL/min consisted of a 20:80 aqueous phase (50 mM ammonium acetate buffer solution with pH 5) and organic phase (acetonitrile), respectively. Then, 20 μL of the sample was injected, and a stationary method of a single mobile phase was applied. All the peaks shown in the chromatogram (Appendix A) were identified and quantified. The retention time of cyclopiazonic acid was 3.68 min.

For the determination of the CPA concentrations in the samples, a calibration curve with seven dilution levels was prepared (concentrations range in between 50 ng/mL–0.025 mg/mL). The linear calibration curve was created by plotting the ratio of the peak area of CPA versus the CPA concentrations in the standards. To reduce the measurement errors, the standard series was measured at least in duplicate.

The chitosan-based film sample preparation for UHPLC took place accordingly: the sample (2 × 2 cm) was placed next to a wall of a 1 mL plastic vial, covered with 1.5 mL of methanol, and the received solution was shaken on a thermoshaker TS-100C (Biosan, Riga, Latvia) for 1 h. Afterward, the film was removed from the methanol, and the excess solvent was evaporated under a stream of N_2_. Received solid matter was restored with 1.5 mL of phosphate buffer (5 mM, pH 2.8). After 10 min of the diffusion process, the solvent sample was subjected to UHPLC. Spiking of film samples (400 µL) with 100 µL of CPA standard solution (0.1 mg/mL) was carried out to ensure the concentration to be within the concentration range used in the calibration curve. The Gouda cheese sample preparation took place according to Zambonin et al. with a slight modification [32] as follows: the cheese sample (0.5 g) was previously cut into small pieces with a knife and weighted into a vial, 1.5 mL of methanol was added and sonicated for 10 min. Reactive methanol was separated from the cheese sample by filtering through the PTFE-20/13 UHPLC filter (Sartorius, Goettingen, Germany) and then evaporated under a stream of N_2_ at ambient conditions (25 °C). Received solid matter was restored with 1.5 mL of phosphate buffer (5 mM, pH 2.8) and then subjected to UHPLC. Recovery calculations were done by spiking cheese samples (400 µL) with 100 µL of CPA standard solution (0.1 mg/mL).

#### 2.6.2. Moisture Content

The moisture analyzer HE 53 (Mettler Toledo, Wien, Austria) at room temperature was used to measure the moisture content (MC) of cheese and films. Cheese samples were analyzed by receiving a cheese piece from cold storage, cutting it into smaller particles, and immediately placing it onto a measuring plate for analysis. The film samples were handled similarly, using scissors for cutting.

#### 2.6.3. pH Value

The pH value of the cheese and film samples were measured at room temperature using a benchtop 781 pH/ion meter (Metrohm AG, Ionenstrasse, Switzerland). The cheese samples were aseptically homogenized in sterile saline solution with a stomacher (Lawson Scientific, Ningbo, China) and analyzed directly. The film pH values were received by measuring the pH of water solvent. For the exact results, the film samples (2 × 2 cm) were immersed into water for two hours and then removed from the water to finalize film activity in the solvent.

#### 2.6.4. Mechanical Properties

Mechanical characterization of chitosan-based films was performed by following the guidelines from the American Society for Testing and Materials (ASTM) D 882 standard method [33]. Rectangular film samples (8 cm × 2 cm) were tested on the Multitest 2.5-i universal testing machine (Mecmesin, Slinfold, UK) equipped with a 100 N load cell, at a crosshead speed of 5mm min^−1^. Tensile strength (TS) was calculated by dividing the load with the average original cross-sectional area in the gage length segment (6 cm) of the sample, while elongation at break (EB) was calculated as the ratio between increased length after breakage and the initial gage length.

#### 2.6.5. Active Properties

The total phenolic content (TPC) of active sachets (~5 mg) was determined by Folin-Ciocalteu’s (FC) phenol reagent according to the protocol outlined in our previous study [34]. Briefly, the small rectangular samples were added into water, followed by the successive addition of FC phenol reagent and aqueous solution of Na_2_CO_3_ (10% *w*/*v*) in the amount of 10 vol% and 20 vol% based on the volume of water, respectively. After the incubation of samples (2 h in dark, 24 °C), the absorbance was measured at 765 nm using the Synergy TM 2 Multi-Detection Microplate Reader (BioTek, Winooski, VT, USA). Gallic acid was used as the standard, and the results were expressed as the mass of gallic acid equivalent (GAE) per mass of the films.

### 2.7. Statistical Analysis

The data were subjected to one-way analysis of variance (ANOVA) with a confidence level of 95% (*p* ≤ 0.05). All the results in triplicate are expressed as the mean ± standard deviation.

## 3. Results and Discussion

### 3.1. Characterization of the Chitosan-Based Films

#### 3.1.1. Mechanical Properties

The mechanical properties of the packaging material used in this study were measured prior to their use as follows: TS (CH) = 6.7 MPa, TS (TA:CH) = 15.0 MPa, TS (CE:CH) = 15.6 MPa, TS (PA:PE) = 27.5 MPa, and EB (CH) = 75.9%, EB (TA:CH) = 28.5%, EB (CE:CH) = 22.9%, EB (PA:PE) = 40.0% (Appendix A). This correlates with what has been shown previously in a comprehensive modeling study by Bajić et al. [14]. Overall statistical difference according to the analysis of variance showed that the films TS property can be considered as different (*p* < 0.05) when active components are added into films, wherein TA and CE showed similar results. In regard to EB, all the films were considered different (*p* < 0.05). Thus, the TS and EB of the biopolymer films were deemed significantly lower (*p* < 0.05) than conventional PA:PE packaging. The values of the tensile strength (TS) and elongation at break (EB) should be adequate when acceptable integrity of good packaging material is requested. The materials produced show strong integrity, considering that heavy food, e.g., fresh pasta, has been packed into similar material and the material withstood the load during the 2 month shelf life [34].

#### 3.1.2. Activity

Along with mechanical properties, the activity of the films was determined and expressed through the total phenolic content (TPC) value. Films used for packing Gouda cheese in this study received TPC values of CH = 0.5 mg_GAE_ g_film_^−1^, TA:CH = 3.2 mg_GAE_ g_film_^−1^, CE:CH = 17.2 mg_GAE_ g_film_^−1^ (Appendix A). Almost six-fold higher (*p* < 0.05) activity by CE was seen and thus indicates the pure compound’s (TA) lack of efficiency as an active component. The higher activity of the CE in the film was expected due to its abundant, diverse phenolic content. Its adverse effect has shown to be even higher than of similar extracts such as oak (9.0 mg_GAE_ g_film_^−1^) and hop (12.7 mg_GAE_ g_film_^−1^) [31,35].

### 3.2. Gouda Cheese Spoilage Microbiota Reduction with the CE:CH Film

#### 3.2.1. Bacteria Reduction with Chitosan-Based Films

The cheese was inoculated with approximately 6.5 log_10_ CFU/g of *E. coli* and wrapped in different types of chitosan films, as seen in Figure 1a. The *E. coli* count on inoculated cheese (iCHE PA:PE), stored at 4 °C, remained throughout the 37 days incubation period at the same value (±0.6 log_10_ CFU/g) as it was at the beginning of the experiment. Between the different tested chitosan films, the *E. coli* count dropped most dramatically with the inoculated chitosan (iCH) film (*p* < 0.05). Accordingly, without the additives (CE or TA), the reduction in *E. coli* count was approximately 2 log_10_ CFU/g. The primary reduction in bacterial count in these samples happened during the first 14 days, while from 14 to 37 days, the *E. coli* count stagnated. In the cheese samples wrapped with inoculated chestnut extract:chitosan (iCE:CH) and inoculated tannic acid:chitosan (iTA:CH) films, the reduction in *E. coli* count was 1 log_10_ CFU/g, and it happened in the first 7 days of the cheese storage.

According to the evidence of Blaiotta et al. [36], Gram-positive lactic acid bacteria (LAB) cell protection could be retrieved when immobilized in CE fiber in an acidic environment. Considering that *E. coli* is Gram-negative, and positively charged chitosan-based materials disrupt the cell membrane [37], it could be considered that CE fiber has the protective effect on the intact cells. Although, it is believed that the effect is not merely related to the fiber. The CE is demonstrated to contain reducing sugars, provides adenosine triphosphate (ATP), and improves the bacteria’s survival [36]. Accordingly, TA:CH films have a similar effect to CE:CH films (Figure 1a), disputing the sugars or fiber effect and indicating the effect of hydrolyzable tannins, and indicating there were destructive effects on the spoilage bacteria, *E. coli*, but only to some extent (until day seven). The lysing result of the neat iCH film stands out in terms of less complexity and its ability to form hydrogen bonds with a higher amount of active cites [37].

The second spoilage bacteria, *P. fluorescens*, was inoculated on cheese in approximately 8.2 log_10_ CFU/g (Figure 1b). The cheese was stored at 4 °C. Conformably, the drop of *P. fluorescens* count was most dramatic with the iCH film, where the count dropped for 6 log_10_ CFU/g, and samples with the iCE:CH and iTA:CH films showed bacteria count reduction for approximately 4 log_10_ CFU/g (*p* < 0.05). Non-inoculated cheese samples were also included in the analysis to exclude the contamination (not shown on graph) and the samples did not show any *E. coli* or *Pseudomonas* spp. presence. Prior the experiment initiation, the count of *P. fluorescens* at inoculation was slightly higher than the *E. coli* count, but it seems that the latter bacteria species is more susceptible to long term storage at low temperatures or the LAB, naturally present in the cheese, was a habitual defense system. The count of *P. fluorescens* dropped even in the samples that were not wrapped in biopolymer films. Herein, one of the main differences between the two bacteria is in their survival conditions. Specifically, *P. fluorescens* is unable to grow under anaerobic conditions, hence the drop. *E. coli*, on the other hand, withstands anaerobic conditions to some extent. 

#### 3.2.2. Fungi Reduction with Chitosan-Based Films

For cheese inoculation with fungi, approximately 4 log_10_ CFU/g of *P. commune* spores were used and incubated at two different temperatures. At refrigeration conditions (4 °C), the drop of *P. commune* count was minimal (less than 1 log_10_ CFU/g) in all the samples regardless of packaging (Figure 2a), which confirms the good survival of the mold spores over time at low temperatures. This shows that the *P. commune* spores are susceptible to components of the chitosan films, temperature conditions, and naturally present LAB only until day 14.

On the other hand, the room temperature conditions (25 °C) affected *P. commune* survival on the cheese samples over time (Figure 2b). After 37 days, the count of *P. commune* had dropped for approximately 2 log_10_ CFU/g on iCHE TA:CH, iCHE CE:CH and iCHE PA:PE samples, while on iCHE CH samples, the *P. commune* count had dropped for only 1 log_10_ CFU/g (*p* < 0.05). Contrary to our study, the study by Duan et al. [28] revealed that the mold count increased (highest at 5.12 log_10_ CFU/g) during 30 day incubation at 10 °C in control (untreated) samples of the cheese, while the count reduction (highest at 1.90 log_10_ CFU/g) was seen when the mozzarella cheese was wrapped in chitosan-based films or coatings. Different species of mold and cheese were used in the study. Evidently, the higher temperature contributes to the film’s active components diffusion process, which has also been shown by Ouattara et al. [38]. Additionally, in a study by Esposito et al. [23], CE, with its polyphenols, was shown to have an inhibiting effect on fungus, both in mycelial and spore form. We also expected that the mold count would increase on cheese wrapped in iCHE PA:PE at room temperature conditions, which are ideal for mold growth. On the contrary, the *P. commune* count dropped during the 37 days long incubation period. Being strictly aerobic mold, most likely, the reason for the reduction is the activity of the LAB. The study of Cheong et al. [39] confirms our theory, where the antifungal effect of LAB was shown on several mold species, *P. commune* being one of them.

#### 3.2.3. Influence of Chitosan-Based Films on Mycotoxin CPA from Cheese

The cheese was inoculated with *P. commune* and repacked into chitosan-based antimicrobial films to retain induced mold growth and secondary metabolite cyclopiazonic acid (CPA) formation. The presence of tannins in the packaging material at a higher temperature (25 °C) hindered the CPA production in cheese samples and had an endorsing effect while held at low temperature (4 °C).

Based on the analysis of variance, which was employed on spiked and non-spiked sample results prior to the subtraction to observe the difference in CPA production, the sample groups were identified as different (*p* < 0.05) while stored in two temperatures (Appendix A).

As seen in Figure 3b, mycotoxin CPA concentration in cheese samples at 25 °C significantly decreased from 6400 (day seven) to 533 µg/kg (day 37) when packed in iCE:CH film, and from 3600 to 1300 µg/kg in iTA:CH film. Furthermore, the values presented are the subtraction of inoculated cheese with the *P.commune* and non-inoculated samples, which gave the direct comparison of the CE and TA effect on growth, eliminating the need for the partitioning between phases.

The cheese packed with iCH film remained constant (200 µg/kg), and the CHE PA:PE sample (without protective film) showed an increase (800 to 1400 µg/kg) in CPA concentration. Likewise, packaging materials themselves depicted opposite outcomes pointing to interaction with CPA and thus possible protective features in regard to food safety (Figure 3).

The mycotoxin concentrations measured from films were low but kept appearing. In the presence of the CE in the film, CPA concentration increased up to 3200 µg/kg (day 37) while with TA, the used UHPLC method enabled the identification of any CPA due to shadowing complex formation with TA (Appendix A). Accordingly, the film without any additives (CH) tended to continue accumulating the mycotoxin, having a CPA concentration of 2200 µg/kg on day 37.

The CPA concentration levels measured in refrigerated cheese (4 °C) elevated from 5900 (day seven) to 8367 µg/kg (day 37), 1867 to 7600 µg/kg, 700 to 5800 µg/kg, and demoted from 1400 to 267 µg/kg when packed in CE:CH, TA:CH, CH and PA:PE packaging, respectively (Figure 3a).

Higher CPA results measured from cheese could be considered as a consequence of tannin chemistry that allows conjugation with CPA (a tetradic indole acid/N-compound) to form stable linkages between carboxylic groups and amines, similarly to the protein–tannin complex reaction [40]. Additionally, the high temperature is advantageous for tannins chemical reactivity [25] and optimal for lactic acid bacteria (LAB) from cheese to be dominant in the microbial competition [41]. Accumulation in CH film at 25 °C may be attributed to the chemical properties of chitosan, possessing more free hydroxyl radicals [42] to continually interact with (Figure 3b). Undetectable CPA in the iTA:CH films is most likely the result of the higher chemical reactivity of tannic acid, but only until a certain limit (being an unvaried chemical) compared to the iCE:CH film, which contains a source of various, more abundant tannin-rich chestnut extract [43]. Furthermore, chitosan-based materials have been observed to have a strong absorption mechanism for various chemical components [44,45], macromolecules [46,47], and also for mycotoxins [48,49], which is used as a safety precaution to conjugate the unwanted particles. This explanation appears to correlate with the specific film activity of this study in terms of mycotoxin. Rapid CPA forming in cheese samples (at 4 °C) is an irrefutable indication of the film’s low activity after packaging, seemingly related to mechanical properties. Cooling storage has a uniforming effect on biopolymers allocation and molecular configuration in materials [50,51]. This is confirmed by the results of inexistent CPA concentration within films on days zero to seven and 37. In time, the chitosan-based chestnut extract film reaches its moisture equilibrium [34], depending on the food product that it is in contact with.

Furthermore, the films potentially act as an absorber in low-temperature conditions, as on day 14 the iCE:CH and iCH film samples depicted higher CPA presence—most certainly influenced by moisture mobility.

Lately, only a few attempts to quantify CPA in cheese have been reported [40]. Comparatively, in white mold cheese, CPA was reported within a range of 1.83–3610 µg/kg measured by HPLC-MS/MS [50,52], and in inoculated cheddar cheese under modified atmosphere (CO_2_ + O_2_) within range of 4–280 µg/kg measured by HPLC [53]. To our knowledge, there are no reports covering CPA quantification in biofilm matrices that have been in contact with food. Although, theoretical multiple mycotoxin absorption by cross-linked chitosan polymer has been reported to be 5.67 g/kg [48]. The results of inoculated gouda cheese (<8367 µg/kg) and chitosan-based films (<3200 µg/kg 2 × 2 cm^2^) remain in the same range reported by Zhao et al. [48] and therefore report non-toxicity in protective films. This is concluded based on the 50% (LD50) lethal dose in rats by oral ingestion (36,000 µg/kg) [54].

### 3.3. pH Value

The overall pH deviation between stored Gouda cheese and acidic chitosan-based films was assessed at every time point to determine the flux direction of acidity known to influence microbial growth. In comparison to the CHE PA:PE/iCHE PA:PE samples, a significant conversion of cheese to become more acidic, while the films altered towards a neutral pH, was observed. All the cheese packed in different chitosan films, conditioned at 25 °C, had a pH decrease from 5.6 to 5.2. The pH of cheese from the iCH film started to decrease right away compared to the results of cheese packed in the iCE:CH and iTA:CH films (Figure 4b,c). The incorporation of CE into chitosan-based film showed not to have an additional pH lowering effect on the cheese compared to the TA effect (*p* = 0.05). A delayed change of pH was determined in the latter packaging’s until day 14 when it started to stabilize. In regard to the iCHE PA:PE cheese samples, a wave-like pH change was seen with a sharp pH increase from 5.6 to 5.8, down to 5.7 by day 14 and up again to 5.9 on day 37.

The cheese pH changed slightly differently when stored at 4 °C. While packed in the TA:CH/iTA:CH film, the pH values always stayed above the values of other biofilm-packed cheese, and did not go lower than pH 5.3, regardless of the spoilage microbes (*p* < 0.05). The influence of the spoilage microbes on the cheese pH could be observed. To be more exact, without spoilage bacteria, it took 7 days for the pH of cheese in the TA:CH film to drop (Figure 4a). With Gram-negative bacteria, a pH drop started to happen after 7 and 14 days, respectively, with *E. coli* and *P. fluorescens* (Figure 4d,e). *P. commune* influenced pH decrease uniformly in each biopackaging throughout the time points (Figure 4f).The iCHE PA:PE sample results were diverse with a higher temperature, although after a similar pH jump, they were strongest with *P. fluorescens* presence up to pH 6.0, and the iCHE PA:PE samples reached a plateau in pH value from day 14.

The pH trend in films (25 °C) took a similar course in every sample set, with a difference in the films containing tannin-additives, being both lower and identical in values (acidic) (*p* < 0.05) than the CH film. The pH of the CH film rose to 5.0 when it stabilized. In the TA:CH and CE:CH films, the pH reached a plateau at 4.8 on average, which was similar in the presence of P. commune (Figure 4b,c). Simultaneously, lower temperature (4 °C) seemed to prevent pH elevation and kept it at the levels of pH 4.4 (blank, *P. fluorescens*, *E. coli*) and 4.6 (*P. commune*) in the CE:CH films. The preventing effect did not apply to the TA:CH film and the pH was recognized as statistically different from the other tannin-rich films (*p* < 0.05).

Based on the CHE PA:PE/iCHE PA:PE results, moisture mobility influence could be considered, and the samples could experience the so-called buffering effect. This reflects matrix degradation, which leads to free water mobility with a result of elevated pH values. In our previous work, we have seen that active components diffusion in between the food matrix and active CH:CE film reaches an equilibrium and thus explains the plateau after day 14. The observation correlates to the studies of other authors, confirming that the chitosan application has the pH altering effect in a different direction and only to a small extent. For example, when the cheese is packed with gelatin:chitosan film, the pH changes from 4.6 to 4.4 [55]. The pH is reported to increase from 5.3 to 5.9 while incorporated in cheese mixture and packed in alginate coating plus a modified atmosphere environment [56]. Saloio cheese’s pH ranged from 4.72 to 5.12 (day 37) when coated with chitosan-based natamycin edible film [57].

The changes in the pH of films are mostly attributed to the electrostatic and hydrophobic interactions between chitosan (cationic) and lipids (anionic) [18,58]. Furthermore, it can be presumed that the warmer environment affects the chitosan network for higher diffusion of active components, while lower temperature keeps the integrity of the material network. Additionally, CE contains fiber, which contributes to the material integrity and thus to a lower diffusion.

### 3.4. Moisture Mobility

The moisture mobility between chitosan-based films and stored Gouda cheese was evaluated for its possible impact on the growth of pathogenic microbes on cheese. Analysis using an automated moisture analyzer was applied to samples initially and at the end of the experiment. Expectantly, the films acted as moisture absorbers, while cheese yielded moisture. The MC in cheese samples, contaminated with two different Gram-negative bacteria, in an anaerobic surrounding at 4 °C, depicts no distinct species that influence associated differences. Temperature’s influence on fungi inoculated cheese, packed in different packaging, stands out with similarity when conditioned at 4 °C and diversity at 25 °C. The CH film without additives performs superior absorbing properties compared to the cheese, while the CE:CH and the TA:CH films, receive results with uniform outcomes, most potentially due to high degradation and intact matrices, respectively (Table 2).

All the samples were packed in additional conventional PA:PE packaging to avoid environmental effects on the MC. However, the CHE PA:PE cheese MC results refer to temperature affected disparities and diversity aspects of biological material after repacking. A slightly lower MC was measured compared to the starting point MC of 33.5 ± 2.0%. Moisture contents on day 37 were the lowest in the CH/iCH packed cheese samples (20.1 ± 2.2%), with results being on average 24.6 ± 1.8% higher in the TA:CH/iTA:CH packed cheese and even higher in the CE:CH/iCE:CH packed cheese (25.6 ± 2.0%) (*p* < 0.05). Likewise, by the same time, the CH/iCH films gained moisture on average up to 62.4 ± 1.5%, the TA:CH/iTA:CH films 51.0 ± 1.6%, and the CE:CH/iCE:CH films 52.2 ± 2.3% (day 37) showing matrix binding similarities in additive-films. Enhanced moisture absorption is most likely attributed to (i) CH: protein interactions and thus lowering the cheese’s protective lipid layer [18], (ii) CE being a starch (good free water binder) incorporator into chitosan-based films [24], and (iii) TA, via its cross-linking mechanism, being able to strengthen the film matrix to be resilient to moisture absorption [59]. Controlling moisture during cheese processing has a technical connotation for final cheese quality [3], and these results show the benefits of CE in regard to moisture absorption, performing absorption to a lower extent and preserving cheese moisture levels time-wise.

### 3.5. Effect of the Film on a Food Safety

In addition to the mechanical integrity, it is mandatory to determine the safety of a new material when it is applied on food. All the ingredients used in films were of natural origin and declared as food safe [30]. A comprehensive study by Hu and Gänzle [60] shows chitosan’s bactericidal effect towards several pathogenic microbes on artificially contaminated intermediate moisture foods, and states that the lethality is limited up to 5 log_10_ CFU/g. Films incorporated with CE in this study are able to inhibit induced spoilage up to 4 log_10_ CFU/g, indicating the efficiency to ensure food safety. One of the goals of this study was get more insight into the fungi development and its mycotoxin production. With an interesting outcome and a positive outcome in regard to food safety, the mycotoxin is absorbed into the film’s matrix, but does not migrate back to the food surface when the storing temperature is 25 °C (Figure 3). This outcome broadens the storage possibilities surrounding temperature range; however, it should be emphasized that this was not the case at 4 °C and further studies should be conducted for the nethermost temperature. Furthermore, food safety is in balance when several food processes (lipolysis, acidity change) are induced. According to our results of moisture content and pH change that correlates to acidity, the films could be considered as promoting food safety, as the changes were minimal.

## 4. Conclusions

Based on the results, it can be concluded that chitosan film enriched with chestnut extract reduces extreme bacterial (up to 6 log_10_ CFU/g) and fungal (up to 4 log_10_ CFU/g) contamination more actively at 25 °C than at 4 °C. The primary decrease in contaminators happened at 14 days and towards *P. fluorescens* most efficiently. The results indicate that the addition of commercial chestnut extract has the equivalent effect of chemical grade tannic acid and chitosan film solely lacks protection—first because of the high degradation. The chestnut extract enriched chitosan film seems to operate as an absorbent of mycotoxin cyclopiazonic acid at 25 °C while lowering the toxicity level in cheese. Although, even in airtight conditions, the cheese yielded moisture to the chestnut extract chitosan film throughout all the samples, on average up to 8 ± 2% of its initial moisture content. This alters the authentic Gouda cheese color to a dark brown (Appendix A). Consequently, packing a high lipid food product—Gouda cheese—into chestnut extract enriched chitosan film could be a good strategy as it influences its pH for only 0.2 units and ensures food safety with active compounds, certainly even longer than 37 days. For further studies, the selected antioxidant and antimicrobial biomarkers extracted from the CE can be used to provide cheese with no visual effects. Additionally, the components chosen for in vivo protection of packed food are entirely safe for use in the food.

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
