# Peer review of "Reduction in Spoilage Microbiota and Cyclopiazonic Acid Mycotoxin with Chestnut Extract Enriched Chitosan Packaging: Stability of Inoculated Gouda Cheese"

_foods, 2020, doi:10.3390/foods9111645_

Round 1

Reviewer 1 Report

General comments:

The objective of the manuscripts is to characterize antimicrobial film based on chitosan and to evaluate its efficacy to preserve semi hard cheese contaminated with specific microorganism. The topic of the manuscripts is interesting but minor revision are required. As general comment I would suggest writing results separate by the discussion. The paper as it is written is not easy to read and it is difficult to follow the discussion related to the hypothesis of the work. Moreover, results from statistical analysis must be described and used to discuss the data.

Discussion about the safety of the film in contact with food must be included.

Following minor revision on the text

Introduction

L35-38: Describe the cheese giving details on water activity that is the most critical parameter for microbiological stability of cheese. The high lipid content have an impact on oxidation.

Material

Please specify if all film constituents are food grade. The chitosan-based film must be used in contact with cheese and due to the nature of the biopolymer material can simply release compound to the food or solubilize by absorbing water from the food. Thus, it is important to use only food grade materials.

2.4 experimental design

Please explain better who the variables are, and the level for each variable. It is not clear from the text. You have written 4:4:2 factorial design: What it means? I can suppose that it means 4 different packaging x 4 microorganism (it is not clear) and 2 temperature, but the total samples must be 32 and in the figure 1 less samples are shown. Please explain better.

Statistical analysis

Why did you analyze data by one-way ANOVA? If you have three factors you have to perform multifactors analysis

Results

L248: you have written “Slight differences are accounted for disparities and the diversity of the biological material”. Please reports results from ANOVA. Are the differences significant? Why the samples are different? Which independent variables have a significant effect?

Paragraph 3.5 sensory perception

Methods are not reported in material and methods section. Please give details on how did you perform sensory analysis? Visual appearance is not scientific data.  Please delate this resulta if not supported by scientific data.

Author Response

General comments:

The objective of the manuscripts is to characterize antimicrobial film based on chitosan and to evaluate its efficacy to preserve semi hard cheese contaminated with specific microorganism. The topic of the manuscripts is interesting but minor revision are required. As general comment I would suggest writing results separate by the discussion. The paper as it is written is not easy to read and it is difficult to follow the discussion related to the hypothesis of the work. Moreover, results from statistical analysis must be described and used to discuss the data.

We thank the reviewer for the comments. The manuscript was reorganized on reviewer’s suggestion to improve text readability. Although the section “Results and discussion” remained joint, we made changes to the text by adjusting the numeric results in the beginning of the paragraphs followed by discussion segment throughout the manuscript. Moreover, the statistical description was added to discuss the data in more detail.

Discussion about the safety of the film in contact with food must be included.

We appreciate this suggestion made by the reviewer. A new section 3.6 Effect of the film on a food safety has been added.

Following minor revision on the text

Introduction

L35-38: Describe the cheese giving details on water activity that is the most critical parameter for microbiological stability of cheese. The high lipid content have an impact on oxidation.

Authors followed reviewer's advice and added details of the water activity found in Introduction.

Material

Please specify if all film constituents are food grade. The chitosan-based film must be used in contact with cheese and due to the nature of the biopolymer material can simply release compound to the food or solubilize by absorbing water from the food. Thus, it is important to use only food grade materials.

All the film components used in the study were of highest purity and could be used for food packaging. A reference [30] has been added .

2.4 experimental design

Please explain better who the variables are, and the level for each variable. It is not clear from the text. You have written 4:4:2 factorial design: What it means? I can suppose that it means 4 different packaging x 4 microorganism (it is not clear) and 2 temperature, but the total samples must be 32 and in the figure 1 less samples are shown. Please explain better.

The factorial design 4:4:2 describes numeric quantity of biopackaging materials (4) throughout four (0, 1, 2, 3) timepoints ran at two different temperatures. A correction has been added to the Section 2.4.

Statistical analysis

Why did you analyze data by one-way ANOVA? If you have three factors you have to perform multi factors analysis

Every dataset went through separate ANOVA analysis and based on the results a discussion was formed. The analysis of the variance test was found to be satisfactory to come to a conclusion. The multifactor analysis was not considered, as the deeper analysis was not the focus of this study.

Results

L248: you have written “Slight differences are accounted for disparities and the diversity of the biological material”. Please reports results from ANOVA. Are the differences significant? Why the samples are different? Which independent variables have a significant effect?

A thorough statistical description with ANOVA has been added to Section 3.1.1. The difference was determined as significant due to the incorporation of active components. The significance was found to be not relevant within the sample groups of active components (chestnut extract film received similar results with tannic acid film).

Paragraph 3.5 sensory perception

Methods are not reported in material and methods section. Please give details on how did you perform sensory analysis? Visual appearance is not scientific data.  Please delate these results if not supported by scientific data.

With a unanimous decision, the authors have decided to remove the section from the main article manuscript and place it as supplementary information situated to the article. Additionally, authors agree with the reviewer that visual appearance needs to be supported with data that was not gathered.

Reviewer 2 Report

General comments

The manuscript describes the effect of chitosan film with chestnut extract in the reduction of specific bacteria and fungi in inoculated cheese. The work is valid, the techniques are appropriate and at state-of-art. However, the work developed and the results must be better presented and discussed. Specific comments are addressed below. The English should be improved.  

Specific comments

Line 50: “mindset of circular economy of reusable or/and degradable” this a rather limited view of the concept of circular economy that included recycling as one of the pillars.

Line 52-54: the sentence is misleading. Although showing good properties and potential, in fact we cannot say that chitosan have been successfully used to substitute synthetic packaging, because it is not used commercially and it is not even approved for use as food contact material.

Line 55-58: the meaning of this sentence is not clear

Line 121-124: was relative humidity of ambient controlled?

Line 139: was the chitosan thermosealable? Hardly to believe!

Lines 147-149: the description of the experimental design is not clear and don’t match with what is presented in scheme 1.

Lines 175 and following: the text needs to be better structured. The reader needs to go through several sentences to realize that the analysis is to be performed in extracts from cheese and from packaging materials.

Lines 207-210: which were the conditions used in the determination?

Lines 240-244:  the authors discuss the requirements regarding tensile strength based on integrity aspects. The idea is valid. However integrity is much more related to seal tightness and integrity, which is a major drawback of these bio materials that do not seal tightly.

Lines 250-256: I believe the discussion on active properties deserve a deeper discussion and possibly a dedicated section.

Lines 259 and all microbiology section:

  • Was the initial count in the inoculated cheese conformed? How?  
  • It should be confirmed and presented correctly what microbial agents were tested at each temperature. Along the text and Figures there are several mistakes (apparently).

Lines 320 and all section 3.2.3: the concentration of CPA in the films and in the cheese is presented in Figure 3 and discussed in the text. However, the mass transfer and the partition of CPA between the two phases (material and cheese) is not taken into account what could led to wrong conclusions. Also, the impact of temperature in the partition should also be considered.

Line 470 and following: the conclusions are not supported by the data. In fact, it seems that chitosan only (without chestnut extract) is more active.  

Author Response

General comments

The manuscript describes the effect of chitosan film with chestnut extract in the reduction of specific bacteria and fungi in inoculated cheese. The work is valid, the techniques are appropriate and at state-of-art. However, the work developed and the results must be better presented and discussed. Specific comments are addressed below. The English should be improved.

The reviewer is deeply thanked for constructive questions and feedback improving the scientific approach we are conducting. We like to bring out that the lines the questions/ comments were stated towards are replaced and highlighted throughout the manuscript. English was additionally checked and improved.

Specific comments

Line 50: “mindset of circular economy of reusable or/and degradable” this a rather limited view of the concept of circular economy that included recycling as one of the pillars.

As a scientists group dealing with biopackagings and their development we are bound to follow the definition of Circular Economy (“A circular economy aims to maintain the value of products, materials and resources for as long as possible by returning them into the product cycle at the end of their use, while minimising the generation of waste.” source: European commission, eurostat). Our group is multifaceted and conducting work on biopolymer films biochemical engineering from marine and plant-based bio-waste (compositions/characterization), application (current study) and biodegradability (directed by other members of the group).

We acknowledge the mislead and rephrased the sentence.

Line 52-54: the sentence is misleading. Although showing good properties and potential, in fact we cannot say that chitosan have been successfully used to substitute synthetic packaging, because it is not used commercially and it is not even approved for use as food contact material. 

Authors share the viewpoint on the materials usage, where they stand in the world of production and actual industrial application. There is much work to be done. Many aspects are necessary to address when coming forth with a new solution or in this case new material for food packaging. Among many, the materials’ large-scale production capabilities and consumers’ acceptability play the major role in the success. Authors of this study play the role in bringing more knowledge to the specialists who could bring the materials on these next levels. Our work is driven by the mind-set that at the time being, the success of the biomaterials is when they are used for short-term storage alone or in combination with synthetic materials.

The study aims to present new data from an application perspective and show food safety of similar material next to multiple authors around the world. The references chosen to support the sentence are the latest review articles on the topic. All of them stating the high potential of the biomaterials and it is time to minimize the processing of the synthetic petroleum-based polymeric films in food technology.

Line 55-58: the meaning of this sentence is not clear

The sentence has been rephrased.

Line 121-124: was relative humidity of ambient controlled?

Yes, the film forming solutions were prepared in a controlled ambient condition.

Line 139: was the chitosan thermosealable? Hardly to believe!

Thermosealing of the chitosan-materials is possible and happens through Maillard reaction between chitosan and reducing sugars and/or glycerol. Latter influences the Maillard reaction of the reducing sugars being the reactive intermediate. Moreover, the chitosan is composed of glycosamine units prone to undergo Maillard reaction. Although, it must be emphasized that chitosan needs to be in combination with a plasticizer (glycerol) to perform thermoplastic properties. Otherwise, it is determined that the chitosan degradation temperature is lower than its melting point. An additional major effect for the seal is based on the bound water in the film matrix, which at appropriate pressures and temperature is responsible to partially dissolving the biofilm and by adjusting, the time and temperature for the seal between both films, the uniform film is being formed. For now, we have found the balance between the components and the method, which have given the films satisfactory integrity and thus sealability.

Lines 147-149: the description of the experimental design is not clear and don’t match with what is presented in scheme 1.

The experimental design has been modified and improved with explanatory sentences.

Lines 175 and following: the text needs to be better structured. The reader needs to go through several sentences to realize that the analysis is to be performed in extracts from cheese and from packaging materials.

We thank the reviewer for detailed perception and with it, the direction to make the text structured. The initial idea of text arrangement covering cheese sample results & discussion followed by film sample results & discussion at 25 °C and the same all over again at 4 °C was changed. Instead, all the cheese and film results covering both temperatures are presented first-hand followed by overall discussion in the same order. Therefore, the section “Results and discussion” was decided to keep joint to keep different data outcomes intact.

Lines 207-210: which were the conditions used in the determination?

Sample preparation was seen through at ambient temperature.

Lines 240-244:  the authors discuss the requirements regarding tensile strength based on integrity aspects. The idea is valid. However, integrity is much more related to seal tightness and integrity, which is a major drawback of these biomaterials that do not seal tightly.

The authors agree with the reviewer and mutually acknowledge the drawback as the sealability of our films for us has remained a continuous study topic even today. Herein, when chitosan films are produced by the casting method, they turn out fragile and thin. The addition of plant extracts increases the thickness and with that, also the internal tension increases. To countersize the tension the plasticisers are added to the composition making them more resilient to the breaking. Based on our observations when working on sealability the tight seal is highly associated with pressure next to temperature and the composition.

Lines 250-256: I believe the discussion on active properties deserve a deeper discussion and possibly a dedicated section.

We appreciate this suggestion made by the reviewer. The “Characterization of the chitosan-based films” paragraph has been reorganized, starting with film's mechanical properties results & discussion followed by film activity results & discussion.

Lines 259 and all microbiology section:

Was the initial count in the inoculated cheese conformed? How?

The initial count was confirmed in the same way as for all the other samples. It is written in the material and methods section that the samples were also analysed on the day 0. The explanatory sentence has been included into text.

It should be confirmed and presented correctly what microbial agents were tested at each temperature. Along with the text and Figures there are several mistakes (apparently).

The entire microbial agent (E. coli, P. fluorescens, P. commune) related results & discussion have been presented in separate sections with temperatures included. It was found that the temperature was missing from P. fluoresence section and added accordingly into the text. Additionally to the text, the experimental design (Scheme 1) represents the sample sets applied to a certain temperature.

Lines 320 and all section 3.2.3: the concentration of CPA in the films and in the cheese is presented in Figure 3 and discussed in the text. However, the mass transfer and the partition of CPA between the two phases (material and cheese) is not taken into account what could led to wrong conclusions. Also, the impact of temperature in the partition should also be considered.

To follow the CPA concentration was done with the intention to further follow the potential effect on the antifungal properties of the film on the food by the migration of the active component from the film. This experiment were support data to the plate count in Figure 2. Furthermore, the values presented are subtraction of inoculated cheese with the P.commune and non-inoculated samples, which gave the direct comparison of the CE and TA effect on growth, eliminating the need to do the partitioning between phases. Moreover, the transport phenomena of the active compounds between the materials will be part of our next studies, where this effect will be in detail examined. In conclusion, we agree with the reviewer of the importance this data.

Line 470 and following: the conclusions are not supported by the data. In fact, it seems that chitosan only (without chestnut extract) is more active. 

Authors agree with the reviewer. On authors, a mutual decision the section was decided to take out from the main manuscript and placed as supplementary information into Appendix A.

Reviewer 3 Report

The paper ‘Reduction of spoilage microbiota and cyclopiazonic acid mycotoxin with chestnut extract enriched chitosan packaging: Stability of inoculated Gouda cheese’ is an interesting study of the influence of active chitosan-based films with chestnut extract, and tannic acid into microbiological spoilage reduction and chemical indices of commercial Gouda cheese during storage. Some improvements are needed.

Major comments:

Authors should add the information about statistical significance between obtained results (not only within one sample, but by adding comparison between samples) through entire manuscript. Unfortunately it is hard to follow discussion and analysis without such information as well as it is hard to confirm justification of conclusions. Such analysis can have influence into discussion, so discussion should be adjusted to the results of statistical analysis (if needed) as well. Statistical significance should be presented on figures and tables in the main text of the manuscript – this can make the paper more readable for potential readers.

Minor comments:

Add reference to Figure 3a.

Author Response

The paper ‘Reduction of spoilage microbiota and cyclopiazonic acid mycotoxin with chestnut extract enriched chitosan packaging: Stability of inoculated Gouda cheese’ is an interesting study of the influence of active chitosan-based films with chestnut extract, and tannic acid into microbiological spoilage reduction and chemical indices of commercial Gouda cheese during storage. Some improvements are needed.

Major comments:

Authors should add the information about statistical significance between obtained results (not only within one sample, but by adding comparison between samples) through entire manuscript. Unfortunately it is hard to follow discussion and analysis without such information as well as it is hard to confirm justification of conclusions. Such analysis can have influence into discussion, so discussion should be adjusted to the results of statistical analysis (if needed) as well. Statistical significance should be presented on figures and tables in the main text of the manuscript – this can make the paper more readable for potential readers.

The manuscript was reorganized at the reviewer’s suggestion to make it easier to follow. The headline “Results and discussion” remained the same. Readability in the section was improved by adjusting the numeric results at the beginning of the paragraphs followed by the discussion segment, and this with every dataset in the manuscript. The statistical description was added throughout the manuscript and could be followed with highlighted locations in the manuscript.

Minor comments:

Add reference to Figure 3a.

The correction has been implemented with lines 353 and 356.

Reviewer 4 Report

The authors evaluated the Reduction of spoilage microbiota and cyclopiazonic acid mycotoxin with chestnut extract enriched chitosan packaging: Stability of inoculated Gouda cheese.

The manuscript deals with interesting topic, but the manuscript requires significant improvement.

Specific remarks:

The phrase in Lines 40-42 “There is a list of cheese contaminators, but few concerning ones that can be named are bacteria Pseudomonas fluorescens and Escherichia coli and fungi Penicillium commune” does not come from reference 4. Yang, Y., Li, G., Wu, D., Liu, J., Li, X., Luo, P., Hu, N., Wang, H., Wu, Y. Recent advances on toxicity and determination methods of mycotoxins in foodstuffs. Trends in Food Science & Technology 2020, 96, 233–252. https://doi.org/10.1016/j.tifs.2019.12.021.Please correct with the right one.

Lines 43-44. You could also add the new publication “Agriopoulou, S.; Stamatelopoulou, E.; Varzakas, T. Advances in Analysis and Detection of Major Mycotoxins in Foods. Foods 2020, 9, 518”

Line 86. Cladosporium cladosporioides. Please correct it in italics.

Author Response

The manuscript is satisfactory and needs some minor corrections. Line 89. Cladosporium cladosporioides. Please correct in italics. Line 276. E.coli. Leave a space. Line 372. cm2. Is it ok? Line 458. The pH The significant difference is identified ... not make sense.

We thank the reviewer for the corrections, which were all accepted and corrected in the manuscript.

Specific remarks:

The phrase in Lines 40-42 “There is a list of cheese contaminators, but few concerning ones that can be named are bacteria Pseudomonas fluorescens and Escherichia coli and fungi Penicillium commune” does not come from reference 4. Yang, Y., Li, G., Wu, D., Liu, J., Li, X., Luo, P., Hu, N., Wang, H., Wu, Y. Recent advances on toxicity and determination methods of mycotoxins in foodstuffs. Trends in Food Science & Technology 2020, 96, 233–252. https://doi.org/10.1016/j.tifs.2019.12.021.Please correct with the right one.

The citation has been checked and corrected. The citation is correct for Penicillum commune but the sentence was misssing additional citation for the bacteria.

Lines 43-44. You could also add the new publication “Agriopoulou, S.; Stamatelopoulou, E.; Varzakas, T. Advances in Analysis and Detection of Major Mycotoxins in Foods. Foods 2020, 9, 518”

Authors thank the reviewer for one of the newest article in the field that widens our understanding of the topic. The reference has been included in the manuscript.

Line 86. Cladosporium cladosporioides. Please correct it in italics.

We thank the reviewer for this notice. The correction has been implemented and the manuscript checked for similar inaccuracies.

Round 2

Reviewer 2 Report

Please see the attachment for the comments.

Author Response

We thank the reviewer for the suggested improvements. Both suggestions have been included in the manuscript.

Reviewer 3 Report

I have no further comments.

Author Response

No comments from the reviewer were given.